# Effect of EVA Polymer and PVA Fiber on the Mechanical Properties of Ultra-High Performance Engineered Cementitious Composites

**DOI:** 10.3390/ma16062414

**Published:** 2023-03-17

**Authors:** Feifei Yan, Peng Zhang, Fang Xu, Wufeiyu Tan

**Affiliations:** Faculty of Engineering, China University of Geosciences, Wuhan 430074, China

**Keywords:** cable channel, ultra-high performance engineered cementitious composites UHP-ECC, EVA polymer, PVA fiber, mechanical properties, SEM micromechanism

## Abstract

In order to study the mechanical properties of ultra-high performance engineered cementitious composites (UHP-ECC) used for cable channel repair, orthogonal tests were carried out with four influencing factors, water binder ratio, silica fume, fly ash and mortar ratio, to obtain the optimum mix ratio of the cement paste. On this basis, the effects of ethylene-vinyl acetate (EVA) polymer and polyvinyl alcohol (PVA) fiber on the fluidity, flexural strength and compressive strength of UHP-ECC were studied, and the micromechanism was analyzed with SEM. The results show that the fluidity of UHP-ECC material prepared was 170–200 mm, which meets the requirements of working performance. The average compressive strength at 28 days reached 85.3 MPa, and the average flexural strength at 28 days reached 22.3 MPa. EVA polymer has a fast film forming rate in an alkaline environment. The formed polymer film wraps the fiber, enhances the bridging role between the fiber and the matrix and increases the viscosity of the material. Therefore, the early flexural strength is significantly improved. The 1-d flexural strength of UHP-ECC material mixed with 9-mm fiber is increased by 18%, and the 1-d flexural strength of 3-mm fiber is increased by 15%. Due to PVA fiber’s high elastic modulus and tensile strength, it improved the flexural and tensile properties of the material after incorporation, especially in the later stages; the 28-d flexural strength of UHP-ECC material mixed with 9-mm fiber increased by 12%, and the 28-d flexural strength of 3-mm fiber increased by 7%. It was concluded that the effect of 9-mm PVA fiber is better than that of 3 mm PVA fiber.

## 1. Introduction

In order to conform to the development process of urbanization and alleviate the current situation of urban land shortage, underground cable transmission is commonly used in cities. The urban cable rate exceeds 50%, and some even reach 75% or more [1]. The underground environment is complex, and structural diseases such as water leakage (water damage), lining crack and lining corrosion often occur during the operation of cable channels [2]. Since various types of cables are mainly placed in the cable channel, the negative impact of structural failure is more prominent than other underground projects [3]. The repair materials need to have self-compactness, quick solidification, early strength, corrosion resistance and other properties to meet the demands of rapid and accurate structural repair.

UHP-ECC consists of cement, fine sand, other admixtures and other substances to form a matrix paste material, adding ethylene-vinyl acetate copolymer (EVA) to increase dispersion and polyvinyl alcohol fiber (PVA) distributed in a disorderly direction. It is a new ultra-high performance engineered cementitious composite with high mechanical properties, meeting the requirements of rapid hardening, early strength and corrosion resistance [4].

At present, the main materials used in underground structure repair are ECC and UHPC. ECC has high ductility, high toughness and high corrosion resistance, but its economic benefit is not high. The low water-cement ratio of UHPC can meet the requirements of self-compactness, quick solidification and early strength, but steel fiber is prone to corrosion in underground cable pipelines, leading to corrosion disease. Therefore, UHP-ECC, with excellent tensile performance, corrosion resistance and leakage resistance, has great application prospects in the repair of underground cable channel diseases.

At present, the following research has been carried out in the UHP-ECC field both at home and abroad. Yao [5] contrasted ordinary reinforced concrete beams (RC) and steel fiber reinforced UHP-ECC beams (RU). The mechanical properties showed that the bearing capacity of the RU beam was close to that of the RC beam when the reinforcement ratio reached 1.86%, which showed that UHP-ECC is feasible instead of the RC beam.

On this basis, UHP-ECC was modified with nano-CaCO3 (NC), and the maximum of friction stress between fiber and matrix was obtained when the NC content was 3% [6]. Yu et al. [7] used recycled fine powder (RFP) to replace 50% cement content in UHP-ECC to achieve a green way to reduce carbon emissions. Cai Weijian [8] studied that the use of UHP-ECC in the circular arc interlayer was beneficial to improve the bearing capacity and ductility of the fiber-reinforced polymer (FRP) confined curvilinearization concrete square column, and its stress-strain curve showed good ductility after the peak value; the UHP-ECC interlayer was beneficial to suppress the lateral expansion of its core concrete. Zhou [9] pointed out that UHP-ECC exhibited high potential for improving the fatigue performance of modern infrastructures. The applications of UHP-ECC above are more about the reinforcement of composite beams, highway pavements and interlayers. So far, few scholars have proposed to use UHP-ECC to repair the diseases of underground cable channels.

Therefore, in order to develop UHP-ECC repair materials with good compatibility with the cable channel concrete structure, excellent performance, convenient construction and economic and environmental protection. This paper first uses the orthogonal method to set up nine groups of cement paste tests to study the effect of the water cement ratio, silica fume, fly ash and mortar ratio on the 7-d strength, test the fluidity and flexural and compressive properties and optimize the cement paste mix ratio; then, UHP-ECC is prepared by mixing EVA polymer and PVA fiber through a cross design, and the mechanical properties of UHP-ECC and cement paste in fluidity and flexural and compressive strength are compared; Finally, the micromechanism that leads to the change in mechanical properties of the material is analyzed with SEM images.

## 2. Test

### 2.1. Performance Indicators

Referring to DL/T5484-2013 code standard [10], the compressive strength of the test block made in this paper was greater than 19 MPa in 1 d, 40 MPa in 3 d and 60 MPa in 28 d. The flexural strength of the test block was greater than 10 MPa in 28 d.

### 2.2. Testing Material

Yadong P II 52.5 cement was selected for this experiment, and its chemical composition is shown in Table 1. Class F fly ash conforms to ASTM C618, and its chemical composition is shown in Table 2. The uncompacted silica fume meets the ASTM C1240 standard, and its chemical composition is shown in Table 3. The density of quartz sand was 2650 kg/m^3^, the maximum particle size was 0.18 mm and the content of SiO_2_ was more than 99.6%. The water-reducing agent adopted a liquid polycarboxylic acid series high-performance water-reducing agent. Organosilicon defoamer was selected as the defoamer, with a content of 0.2% [11]. The accelerator was an aluminum-oxide clinker powder accelerator containing soda ash, with a fineness of less than 12.6%. EVA polymer with an average particle size of 80–100 μm was used; its use in UHP-ECC is to improve liquidity, and its physical properties are shown in Table 4. Adding fiber into UHP-ECC can improve the crack resistance, toughness and tensile strength. Two different lengths of PVA high-strength and high elasticity modulus fibers were adopted, 3 mm and 9 mm; the fiber diameter was 40 μm. Tensile strength could reach 1400–1600 MPa and the apparent density was 0.91 g/cm^3^; other physical properties are shown in Table 5.

### 2.3. Test Methods and Experimental Equipment

The cement paste performance test method refers to GB/T17671–2021 [12], and the UHP-ECC performance test method refers to ASTM C1116/C1116M-2010a [13]. The main experimental equipment and apparatus for this experiment are shown in Table 6. Test according to the matching ratio, weigh the required dry materials, weigh the cement, quartz sand, fly ash, silica fume powder, EVA polymer and accelerator into the cement mortar mixer and mix well; then, add the liquid combined with water, defoamer and water reducing agent, slow stir the mixture for 1 min and fast stir for 3 min. After mixing, put part of the mortar immediately into the cement fluidity electric jumping table, test fluidity, select the final test formula to meet the fluidity between 170–200 mm; the other part is poured into the cement mortar 40 × 40 × 160 test molds and plastered after vibrating on the cement mortar shaker for 1 min. Prepare the test block of a size 40 mm × 40 mm × 160 mm for compressive and flexural strength tests. The poured test block is then cured under the standard room temperature environment, and use the Digital Mortar Setting Time Tester to measure the setting time of the concrete mortar block. After 24 h, demold it and immediately move it into a concrete standard curing box at a temperature of (20 ± 2) °C and a relative humidity of more than 95% until the specified test age. Use a microcomputer-controlled electronic bending testing machine and microcomputer control electro-hydraulic servo pressure testing machine to test compressive and flexural strength.

## 3. Mix Ratio Design

### 3.1. Orthogonal Test Design of Cement Paste Ratio

The excellent performance of cement paste is the basis for the preparation of UHP-ECC, this paper uses the orthogonal test method [14] to determine the optimal cement paste mix ratio, considering the mechanism of each component, and the mix ratio that meets the requirements of JGJ55-2011 [15] is designed. The water-cement ratio [16] (factor A), silica fume powder [17] (factor B), fly ash [18] (factor C) and colloidal sand ratio [19] (factor D), as the four material components influencing factors, and the level parameter setting of each factor and the maximum influence factor of 7-d strength [20], are shown in Table 7.

Figure 1 is the index comparison chart of the impact factor. As shown in Figure 1, the three points of factor A have the largest span in height, so the impact on the strength of 7-d strength is the largest, while the points of factor B, C and D have relatively small difference in height, indicating that the impact on the 7-d strength is small, and it is a secondary factor. From the figure, the maximum of influence factor A is A_1_, the maximum of B is B_3_, the maximum of C is C_2_ and the maximum of D is D_1_; therefore, the optimal production conditions are set at A_1_B_3_C_2_D_1_. A_1_ is 0.25 water-cement ratio, B_3_ is 12% silica fume ratio, C_2_ is 10% fly ash ratio and D_1_ is 1.1 cement-sand ratio, which is the optimal cement paste mix ratio under this condition. 

### 3.2. UHP-ECC Mix Ratio Cross-Test Design

On the basis of the cement paste test, EVA polymer and PVA fiber were mixed to form UHP-ECC, and the mix ratio parameters were set as follows:

The total amount of cementitious material 1010 kg/m^3^, silica fume and fly ash composite admixture accounted for 22% of the mass of cementitious material. The cement-sand ratio was 1.1; the specific mix ratio is shown in Table 8. The percentage of water reducing agent in UHP-ECC, numbered B1 to B4 in the cementitious material, was 1%, 1.1%, 1.2% and 1.5%, respectively. Compared with the 3 mm and 9 mm fibers without EVA polymer, the percentage of fibers in UHP-ECC numbered C1 to C2 in the cementitious material is 1%. The percentage of 3-mm and 9-mm fibers mixed with 5% EVA polymer in UHP-ECC numbered D1 to D2 in the cementitious material was 1%. The percentage of modified 3-mm and 9-mm fibers mixed with 5% EVA polymer in UHP-ECC numbered E1 to E2 in the cementitious material was still 1%. The number B0 was the cement paste ratio, and the UHP-ECC material number C0 had no fiber added.

## 4. Results and Discussion

### 4.1. Effect of EVA Polymer

For UHP-ECC, the incorporation of EVA polymer can improve the rheological properties between the fibers and the matrix [21]. Table 9 shows the working and mechanical property data of UHP-ECC prepared in this paper, and Figure 2 shows the comparison curve of mechanical properties before and after adding EVA polymer; a is the flexural strength and b is the compressive strength.

Adding EVA polymer can ensure that the fluidity of UHP-ECC reaches 170–200 mm. At the same time, as shown in Figure 2a, the flexural strength of the material in 1 d is higher than that of the no EVA polymer material. The flexural strength of UHP-ECC material mixed with 9-mm fiber is 18% higher than that of the material without EVA, and the flexural strength of 3-mm fiber in 1 d is 15% higher. Figure 2b shows that the compressive strength of UHP-ECC material decreases after EVA is added, and the compressive strength of D1 and D2 is always lower than C1 and C2. The 1-d compressive strength of UHP-ECC material mixed with 9-mm fiber is 36% lower than that without EVA, and the 1-d compressive strength of 3-mm fiber is 24% lower. The 28-d compressive strength is reduced by 7%, and the 1-d compressive strength of 3-mm fiber is reduced by 15%. The final 28-d minimum compressive strength can reach 79.1 MPa.

The film-forming speed of EVA polymer in cement solution is faster than that in water solution, because the alkaline pore solution destroys the protective colloid polyvinyl alcohol on the particle surface, thus accelerating the fusion stage of particle fracture and cementing materials. Therefore, the flexural strength of the material without EVA polymer early in 1 d is significantly higher than that of the material without EVA polymer, and with the increase in curing time, the flexural strength is gradually lower than that of the material without EVA polymer. The first specific reason for this is that the hydration process of cement will increase the local suction on the particle surface, promote the fusion of polymer particles nearby to form a film and some polymer particles and cement particles form a capsule structure, resulting in the hydration delay or gradual termination of the wrapped cement particles. Second, it can be explained by the “centroplasm theory” [16,22]. Compared with cementing materials and quartz sand, EVA polymer is an elastic medium, which can absorb and consume a large amount of strain energy to achieve energy conversion. EVA polymer is also a soft material, distributed in three-dimensional reticular cement-based materials, forming a flexible centripetal different from sand, and the strength of flexible materials is weak. This is also a good explanation for the fact that the flexural strength is gradually lower than that of non-EVA polymer materials in the later stage.

The reasons for the reduction of compressive strength after EVA polymer is added are analyzed in the following section. First, the interfacial transition zone between EVA polymer and cementitious material is the weak link of cement-based material [23]. With the increase in curing time, the weakening effect of the interfacial is gradually strengthened. When the weakening effect of the interfacial exceeds the toughening effect of EVA polymer, once cracks occur during the pressure bearing process, the cement matrix will rapidly develop around the EVA polymer, leading to the overall compression failure of the material reaching the material limit state in advance. Second, after adding EVA polymer, while improving the dispersion of other materials, it also increases the gap between quartz sand and cementitious materials, resulting in bubbles easily produced during mold installation. In long-term curing, the hydration of bubbles increases, which will reduce the overall strength of materials to a certain extent when under pressure.

### 4.2. Effect of PVA Fiber

Applying the modified material to the pavement coating significantly improves the mechanical properties [24]. Then, by applying modified PVA fiber to UHP-ECC, it can be concluded from the data that the flexural strength is significantly improved. The specific reason is that the modified material can significantly enhance the dispersion ability between fiber and cement, better combine the matrix and fiber material, increase the interfacial adhesion, and thus, enhance the overall flexural strength of the material. The modified PVA fiber matrix is an ideal material for underground infrastructure reconstruction [25].

The reason for the reduction in compressive strength is that the use of modified materials in the process of modifying the fibers, the −CH_3_ on the surface of the modified material, reduces the surface free energy of the fibers and builds up a nano-surface rough structure on the fiber surface by agglomeration [26], and the same proportion of 3-mm fibers in contact with the modified material have a much larger surface area than 9-mm fibers, so the 3-mm fiber agglomeration phenomenon is more obvious, reducing the compressive strength of the material; because the agglomeration phenomenon is obvious, so the 3-mm fiber flexural strength enhancement effect is also lower than the 9-mm fiber (Figure 3).

## 5. Microstructure and Mechanism Analysis

Figure 4a shows the 7-d micro-crack morphology of UHP-ECC mixed with EVA polymer. It can be observed that the surface of the lamellar hydrates C_3_A·3CaSO_4_·32H_2_O is covered with a layer of polymer film. The polymer film was formed at the pre-hydration stage and stabilized after 7 d. It is a kind of interfacial tackifier that can increase the viscosity between the components. The polymer film still plays a cementing role with the components after the cracks are generated. This also fully explains that the flexural strength tends to steadily increase after adding EVA polymer for 7 d, which is larger than the flexural strength before adding EVA polymer.

Figure 4b shows the internal morphology of the hole formed after adding EVA polymer. It can be observed that, even inside the hole, the material still presents a tight and flat state. There are many lamellar substances wrapped in polymer film stacked layer by layer, and the components of the material are tightly bonded to each other, which is also a reason for the overall densification of the material. However, the elastic modulus, strength and stiffness of the polymer itself are relatively low, so the compressive strength decreases after adding EVA polymer.

Figure 5 is the SEM diagram of the fracture of a fiber in the test block E2. Figure 5a shows the fracture state of the PVA fiber; the shape of the fracture surface is irregular, this is because PVA fiber belongs to hydrophilic material, and there is significant chemical bonding with cement-based material during use. The interface between the fiber and the matrix is too strong; when the fiber is under pressure, the fiber is not pulled out, but up to failure, and the number of cracks decreases and the strain value decreases [27]. Therefore, surface modification treatment of PVA fibers is needed to weaken their chemical bonding, which causes fiber pullout and stress transfer to continuously occur during the action of fibers and matrix, and a stable multi-slit cracking process occurs to increase the strain capacity. Figure 5b shows that there are many adhesive materials wrapped by polymer film on the surface of the fiber, and there are dense hydrates and scratches on the surface of the fiber, which indicates that the polymer promotes the bonding of PVA fiber and the cement matrix, so that the material has a better bonding interface, and the force between the fiber and the matrix as a whole transfers, giving better support to the bridging role of the fiber [28]. Therefore, the material has a better bonding interface and the fibers and the matrix as a whole transfer forces to each other and better play the role of fiber bridging [28], which is consistent with the test results of significantly improved flexural strength. In addition, it can also be clearly seen in Figure 5 that the fibers are distributed in random directions, and there is a phenomenon of mutual agglomeration, which is the reason why the compressive strength decreases after adding fibers [29].

In summary, the UHP-ECC developed and prepared in this paper, because of the admixture of EVA polymer and PVA fiber, can effectively prevent water leakage (water damage), lining cracks, lining corrosion and other structural diseases in the cable channel, and can be well compatible with the concrete structure of the cable channel. It has excellent mechanical properties, convenient construction and economic and environmental protection.

In this paper, the corresponding laws were only derived from the experimental data and microscopic phenomena. However, Lee [30] investigated that the proposed model generally well represented the bond behavior of PVA fibers. Jahangir [31] proposed a new analytical model to estimate the SRP-concrete bond strength using a genetic algorithm, which outperformed 22 existing FRP-concrete bond strength models. Li [32] proposed ECC micromechanics-related material microstructures to the composite tensile behavior, which enabled prediction/modelling of ECC tensile behavior. Therefore, there is still a problem in this paper pertaining to the fact that the evolution of the micro-component change process has not been carried out by microscopic simulation, and the experimental data has not been further verified by the analytical model. The related issues need further research and advancement, but the development and application of UHP-ECC materials is foreseeable and worthy of expectation.

## 6. Conclusions

(1)The composition optimization design of UHP-ECC composite matrix material was carried out by orthogonal testing. The optimized ratio was: water-binder ratio 0.25, cement-sand ratio 1.1, silica fume 12%, fly ash 10%, EVA polymer 2%, PVA fiber 1%. An UHP-ECC that could meet the needs of cable channel repair was prepared.(2)The incorporation of EVA polymer could meet the requirements of the material’s fluidity of 170–200 mm and reduce the brittleness of the material. At the same time, the early flexural strength of UHP-ECC material was also significantly improved. The 1-d flexural strength of 9-mm fiber was increased by 18%, the 1-d flexural strength of 3-mm fiber was increased by 15% and the later strength could also meet the standard. The 28-d average flexural strength could reach 20 MPa, and the average compressive strength could reach 80 MPa.(3)The modified fibers were incorporated to significantly improve the flexural strength. The modified 9-mm PVA fibers increased the overall flexural strength of the material by 17%, the flexural ratio by 33% and the maximum flexural strength reached 24.5 MPa; the modified 3-mm PVA fibers increased the overall flexural strength of the material by 10%, the flexural ratio by 19% and the average flexural strength could reach 23.6 MPa. The modification effect of 9-mm PVA fiber was relatively better.(4)Finally, according to the SEM diagram of UHP-ECC, it was analyzed that the incorporation of EVA polymer formed a polymer film to enhance the viscosity between the material components, and the incorporation of modified PVA fiber played a key role in bridging and bonding, so the flexural strength of UHP-ECC material was significantly improved.

## Figures and Tables

**Figure 1 materials-16-02414-f001:**
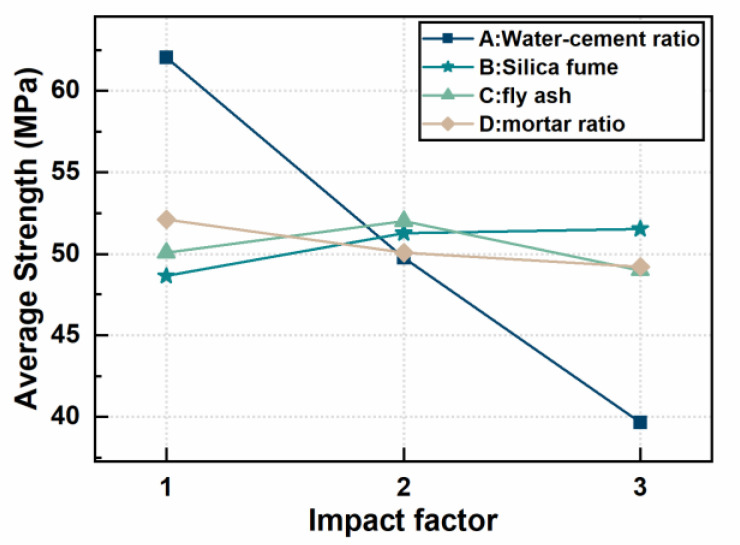
Comparison chart of three indicators of four impact factors.

**Figure 2 materials-16-02414-f002:**
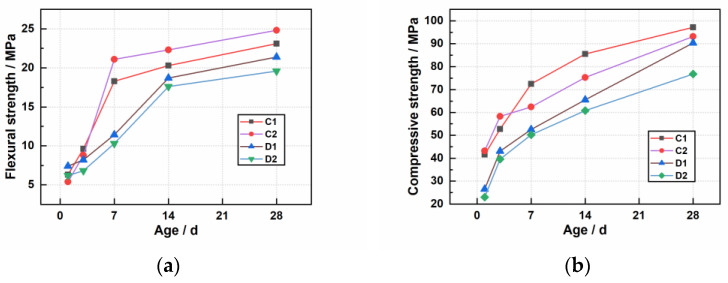
The effect of EVA of the material on the strength: (**a**) flexural strength; (**b**) compressive strength.

**Figure 3 materials-16-02414-f003:**
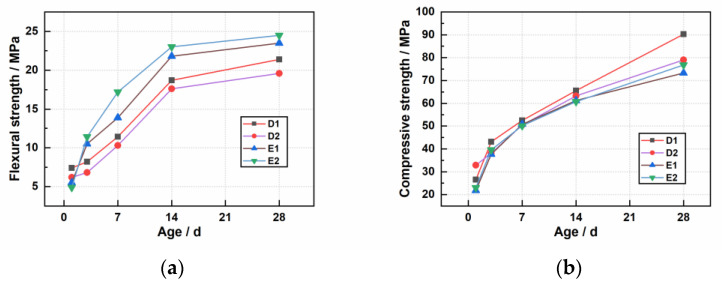
The effect of modified fibers of the material on the strength: (**a**) flexural strength; (**b**) compressive strength.

**Figure 4 materials-16-02414-f004:**
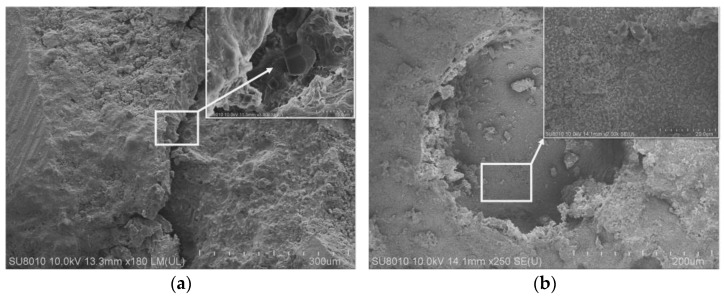
SEM image of joining EVA. (**a**) is the “micro-crack morphology”; (**b**) is the “internal morphology of the hole”.

**Figure 5 materials-16-02414-f005:**
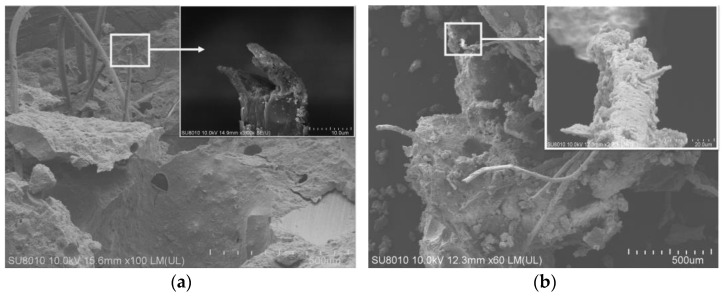
SEM image of joining EVA. (**a**) is the fracture state of the PVA fiber; (**b**) is the “polymer film on fiber surface”.

**Table 1 materials-16-02414-t001:** Chemical composition of cement.

Composition	SiO_2_	Al_2_O_3_	Fe_2_O_3_	CaO	K_2_O	SO_3_	MgO	Na_2_O	
content/%	21.38	4.59	3.13	63.98	0.46	2.17	2.34	0.11	

**Table 2 materials-16-02414-t002:** Chemical composition of fly ash.

Composition	SiO_2_	Al_2_O_3_	Fe_2_O_3_	CaO	K_2_O	TiO_2_	MgO	Na_2_O
content/%	53.97	31.15	4.16	4.01	2.03	1.13	1.01	0.89

**Table 3 materials-16-02414-t003:** Chemical composition of silica fume.

Composition	SiO_2_	MgO	Fe_2_O_3_	CaO	K_2_O	MnO	Na_2_O
content/%	97.91	0.14	0.04	0.29	0.30	0.02	0.21

**Table 4 materials-16-02414-t004:** Physical properties about the EVA polymer.

Physical Properties	Parameter
Color	white
Solid content	≥99%
Ash content (1000 °C)	112 ± 2%
Bulk density	450–600 g/L
Average particle size	≥80 μm
Ph	5–8

**Table 5 materials-16-02414-t005:** Physical properties of the PVA fiber.

Test Items	Standard Date	Measured Date
Linear density (tex)	2400 ± 120	2399
Breaking strength (N/tex)	≥0.40	0.45
Tensile strength (MPa)	≥2000	2.63 × 10^3^
Tensile modulus of elasticity (%)	≥85	88.9
Elongation at break (%)	≥2.5	2.99

**Table 6 materials-16-02414-t006:** The main equipment and instruments of the experiment.

Equipment	Model
Cement mortar mixer	JJ-5
Cement mortar shaker	ZT-96
Cement mortar test mold	40 × 40 × 160
Concrete standard curing box	HY-40B
Microcomputer Controlled Electronic Bending Testing Machine	YDW-10
Microcomputer control electro-hydraulic servo pressure testing machine	YAW4605
Digital Mortar Setting Time Tester	SN—100
Cement fluidity electric jumping table	NDL-3
Other measuring tools	/

**Table 7 materials-16-02414-t007:** Mix ratio of cement past.

No.	Water-Cement Ratio	Silica Fume/%	Fly Ash/%	Mortar Ratio	7 d Compression/MPa
A1	0.25	8	8	1.1	82.08
A2	0.25	10	10	1.2	78.19
A3	0.25	12	12	1.3	76.06
A4	0.30	8	10	1.3	67.66
A5	0.30	10	12	1.1	64.05
A6	0.30	12	8	1.2	57.34
A7	0.35	8	12	1.2	49.67
A8	0.35	10	8	1.3	50.07
A9	0.35	12	10	1.1	48.97

**Table 8 materials-16-02414-t008:** Mix ratio of UHP-ECC.

No.	Water Reducer/g	EVA Polymer/g	9 mm PVA Fiber/g	3 mm PVA Fiber/g	Water/g
B0	10	0	0	0	252.5
B1	10	39.5	7.9	0	252.5
B2	11	39.5	7.9	0	252.5
B3	12	39.5	7.9	0	252.5
B4	15	39.5	7.9	0	252.5
C0	11	39.5	0	0	252.5
C1	11	0	7.9	0	252.5
C2	11	0	0	7.9	252.5
D1	11	39.5	7.9	0	252.5
D2	11	39.5	0	7.9	252.5
E1	11	39.5	7.9	0	252.5
E2	11	39.5	0	7.9	252.5

**Table 9 materials-16-02414-t009:** Performance data of UHP-ECC.

No.	Working Performance	Compressive Strength/MPa			Flexural Strength/MPa			Flexural Compressive Ratio		
	Fluidity	1 d	7 d	28 d	1 d	7 d	28 d	1 d	7 d	28 d
B0	180	40.5	59.6	93.7	6.9	18.2	23.8	0.17	0.31	0.25
B1	165	47.3	53.1	92.6	7.2	11.8	19.2	0.15	0.22	0.21
B2	185	43.7	54.3	90.1	6.9	15.2	18.6	0.16	0.28	0.21
B3	215	30.67	52.8	87.6	6.7	12.1	16.1	0.22	0.23	0.18
B4	255	29.7	47.5	80.5	5.8	11.3	15.3	0.20	0.24	0.19
C0	190	36.1	48.6	80.7	7.3	12.7	19.7	0.20	0.26	0.24
C1	175	41.6	72.5	97.2	6.3	18.3	23.1	0.15	0.25	0.24
C2	177	43.2	62.5	93.2	5.4	21.1	24.8	0.13	0.34	0.27
D1	188	26.5	52.6	90.3	7.4	11.4	21.4	0.28	0.22	0.24
D2	185	32.9	50.4	79.1	6.2	10.3	19.6	0.19	0.20	0.25
E1	185	21.8	50.8	73.4	5.5	13.9	23.6	0.25	0.27	0.32
E2	187	23	50.17	76.8	4.9	17.2	24.5	0.21	0.34	0.32

## Data Availability

Some or all data that support the findings of this study are available from the corresponding author upon reasonable request.

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
