# Peer review of "Effect of EVA Polymer and PVA Fiber on the Mechanical Properties of Ultra-High Performance Engineered Cementitious Composites"

_materials, 2023, doi:10.3390/ma16062414_

Round 1
Reviewer 1 Report
Nil

Reviewer 2 Report
In this manuscript, the effect of EVA Polymer and PVA Fiber on the mechanical properties of UHP cementitious composites were evaluated experimentally. Although the issue is interesting, the presentation of the manuscript needs major revision. Some suggestions are provided below:
- The abbreviations such as EVA, and PVA should be introduced as the first time appear in the manuscript text.
- More exact quantitative results can be added to the last sentences of the abstract.
- There are many previous research works which worked on fiber reinforced cementitious composites and can be added to the literature review. Some of the related publications provided below:
· Lee, D., Lee, S. C., & Yoo, S. W. (2023). Bond Behavior of Steel Rebar Embedded in Cementitious Composites Containing Polyvinyl Alcohol (PVA) Fibers and Carbon Nanotubes (CNTs). Polymers, 15(4), 884.
· Jahangir, H., Rezazadeh Eidgahee, D., & Esfahani, M. R. (2022). Bond strength characterization and estimation of steel fibre reinforced polymer-concrete composites. Steel and Composite Structures, 44.
· Li, J., Qiu, J., Weng, J., & Yang, E. H. (2023). Micromechanics of engineered cementitious composites (ECC): A critical review and new insights. Construction and Building Materials, 362, 129765.
- It is urged to add figures from utilized materials and fibers in the current study.
- Figure 1 is hard to read.
- Figures 2 and 3 can be presented with more resolution.
- More explanations and comparisons needed to be added to the section 4.
- The conclusion can be summarized.
Author Response
Firstly, thank the reviewers for their helpful comments to our manuscript. We have revised the manuscript according to reviewers' comments. In our point-by-point response attached below, reviewer comments are in blue fonts and our responses are in red fonts.
1.The abbreviations of EVA and PVA have been added to the summary of the first occurrence, you also can see lines 11,12 for details.
2.My modification in this part is to delete the scope of the previous test conclusion and add a quantitative description of the test data in the summary. For details, see lines 16-24 in the article.
3.We have made changes and added some references in the paper. Please see lines 281-283, 283-285,285-287 and 380-385for no.30,31,32。
4.We have made changes, the physical properties of EVA polymer and PVA fiber are presented in the form of tables, you can see in line 102 table4 for EVA Polymer’s physical properties, and in line 103 table5 for PVA Fiber’s physical properties, EVA Polymer’s impact for UHP-ECC is to improve liquidity, adding PVA fiber into UHP-ECC can improve the crack resistance, toughness and tensile strength., their impact discussed in article lines 93-95 for details.
5.We have made changes in the Fig1, it represents the index comparison chart of the impact factor. factor A(water-cement ratio) have the largest span in height, and in abscissa,the largest is 1,so in the paper ,we write the maximum of influence factor A is A1, the maximum of B(Silica fume) is B3, the maximum of C(fly ash) is C2, and the maximum of D(mortar ratio) is D1, so the optimal production conditions are set at A1B3C2D1.
6.The quality of figures has been improved. Please see Figs.2, 3 in the article lines198 and 244 for details.
7.Thank you for your suggestion. The section 4 have added some explanations and comparisons in lines 172-181 and 185-202.
8.In the conclusion part, we have modified the order in the discussion of polymer and fiber, and added a lot of essential information basic experimental data in the re-description process. For details, see lines 296-301 in the article.
Round 2
Reviewer 2 Report
The revised version of the manuscript responded all questions and applied all suggestions.